# Knowledge mobilisation of rapid evidence reviews to inform health and social care policy and practice in a public health emergency: Appraisal of the Wales COVID-19 Evidence Centre processes and impact, 2021–23

Micaela Gal[1]*, Alison Cooper[1,2], Natalie Joseph-Williams[1,3], Elizabeth Doe[1], Ruth Lewis[4], Rebecca Jane Law[5], Sally Anstey[1], Nathan Davies[1], Amy Walters[6‡], Robert Orford[2‡], Brendan Collins[2‡], Lisa Trigg[7‡], Chris Roberts[8‡], Sarah Meredith[2‡], Steven Macey[9‡], Andrew Carson-Stevens[2], Jane Greenwell[10‡], Ffion Coomber[2‡], Adrian Edwards[1,2]

1 Health and Care Research Wales Evidence Centre, Cardiff University, Cardiff, United Kingdom, 2 Science Evidence Advice Division, Welsh Government, Cardiff, United Kingdom, 3 Division of Population Medicine, Cardiff University, Cardiff, United Kingdom, 4 Health and Care Research Wales Evidence Centre, Bangor University, Bangor, United Kingdom, 5 Science Team, Health and Social Services Division, Welsh Government, Cardiff, United Kingdom, 6 Health and Care Research Wales, Cardiff, United Kingdom, 7 Social Care Wales, Cardiff, United Kingdom, 8 Knowledge Analytical Services, Welsh Government, Cardiff, United Kingdom, 9 Equality, Poverty and Childrens Evidence and Support Division, Welsh Government, Cardiff, United Kingdom, 10 Information Technology, Cardiff University, Cardiff, United Kingdom

☉ These authors contributed equally to this work.
‡ AW, RO, BC, LT, CR, SM, SM, JG and FC also contributed equally to this work.
* GalM@cardiff.ac.uk

## Abstract

### Background

The Wales COVID-19 Evidence Centre (WCEC) was established from 2021–23 to ensure that the latest coronavirus (COVID-19) relevant research evidence was readily available to inform health and social care policy and practice decision-makers. Although decisions need to be evidence-based, ensuring that accessible and relevant research evidence is available to decision-makers is challenging, especially in a rapidly evolving pandemic environment when timeframes for decision-making are days or weeks rather than months or years. We set up knowledge mobilisation processes to bridge the gap between evidence review and informing decisions, making sure that the right information reaches the right people at the right time.

### Aims and objectives

To describe the knowledge mobilisation processes used by the WCEC, evaluate the impact of the WCEC rapid evidence reviews, and share lessons learned.

**Data Availability Statement:** The anonymised stakeholder survey data (questions and responses (answer selection and free text) has now been made avaialble on the repository Figshare (https://figshare.com). The data file can be downloaded here: https://figshare.com/articles/dataset/Wales_COVID-19_Evidence_Centre_2021-2023_Stakeholder_survey/25912459 Some data and identifiers were removed before adding the data to Figshare (i.e. names. job title, place of work, comments that may identify participants, feedback where participants did not give permission to quote this). This was done to ensure the data was anonymised, as respondents had not consented to have this data shared without anonymisation.

**Funding:** The following authors; MG, AC, ED, NJW, RL, JG, and AE, received salary funding for this work as part of the Wales COVID-19 Evidence Centre (https://healthandcareresearchwales.org/about-research-community/wales-covid-19-evidence-centre), which was funded by Welsh Government through Health and Care Research Wales (https://healthandcareresearchwales.org/), project identification number 522128. Authors SA and ND received reimbursement for their time from the same award, as members of the Evidence Centre public partnership group. The funders had no role in study design, data collection and analysis, decision to publish, or preparation of the manuscript.

**Competing interests:** I have read the journal's policy and the authors of this manuscript have the following competing interests: Authors RJL, AW, RO, BC, LT, CR, SM and SM are employed by Welsh Government. Author AW is funded by Welsh Government through Health and Care Research Wales.

## Methods

Our knowledge mobilisation methods were flexible and tailored to meet stakeholders' needs. They included stakeholder co-production in our rapid evidence review processes, stakeholder-informed and participatory knowledge mobilisation, wider dissemination of outputs and associated activities including public engagement, capacity building and sharing of methodologies. Feedback on processes and evidence of impact was collected via stakeholder engagement and a stakeholder survey.

## Results

Findings indicate that knowledge mobilisation processes successfully enabled use of the WCEC's rapid evidence reviews to inform policy and practice decision-makers during the COVID-19 pandemic in Wales. Realising actual public and patient benefit from this 'pathway to impact' work will take further time and resources.

## Discussion and conclusion

The WCEC knowledge mobilisation processes successfully supported co-production and use of rapid evidence review findings by scientific advisors and policy and practice decision-makers during the COVID-19 pandemic. Identified barriers and facilitators are of potential relevance to wider evidence initiatives, for setting up similar Centres during crisis situations, and supporting future evidence-based policy and practice decision-making.

## 1.0 Introduction

The COVID-19 pandemic dramatically changed health and social care needs, and the way essential services were delivered in Wales and beyond. Health and care policy and practice decisions had to be rapidly made and informed by evidence. The need for evidence-informed policy and practice decisions is undisputed [1]. However, ensuring that the best research evidence is available, accessible and timely is challenging at the best of times, and even more so in the context of a rapidly evolving public health emergency [2]. The pandemic has resulted in some valuable examples of exemplary practices by clinicians and researchers. These should serve as a valuable resource and lesson for managing future crisis situations, and to inform future health and social care policy and practice [3].

Knowledge mobilisation enables research evidence to be understood and used by health and social care policy and practice decision makers, and to make a difference to patients and society. The process moves beyond ending a research project with conference presentation and a peer-reviewed journal publication. The National Institute of Health and Care Research (NIHR) defines knowledge mobilisation as *'Getting the right information to the right people in the right format at the right time, so as to influence decision-making'*, and has a range of resources, which aim to help researchers with knowledge mobilisation of research findings [4]. Knowledge mobilisation is a dynamic and iterative process that includes engagement, co-production, shared learning, dissemination, communication, and the exchange and use of knowledge. Several frameworks, theories, models and guides for mobilising knowledge have been developed. These include the widely used and adapted 'Knowledge to Action' Framework, Developing Evidence Enriched Practice, the 'What Works Networks', and the 'Bridge Building Model' [5]. The latter two of these being targeted towards the use of research evidence in the policy context [6–9]. In 2021, Social Care Wales (a Welsh Government-Sponsored Body)

published its model for knowledge mobilisation, setting out its aims and approach for 'Evidence-enriched practice, planning and policymaking for social care in Wales' [10].

There is much published research on the enablers and barriers to the use of research evidence. In the policy making and health and social care practice context, common barriers include a lack of co-production to ensure that the research is actually relevant and needed, inaccessibility of the evidence (scientific language and findings only available in journals), and failing to meet the timeframe of the policy cycle (research can take years and policy makers need information more rapidly), or practice need (e.g. the clinical pathway has not been considered) [1, 11–13]. Conversely, enablers include early engagement and co-production of research together with end users like policy makers, social care workers, clinicians and members of the public. This helps researchers to understand the need, setting and timeframe for evidence uptake, and makes research findings more likely to be used. It should be noted that research evidence will only partly contribute towards the knowledge used by decision-makers, who will also use advice from domain experts, from those with lived experience, and their own knowledge to inform decisions [14]. In addition, the values and preferences of individuals and communities, and available human and financial resources also play an important part in decision making.

Achieving impact from research is supported by knowledge mobilisation processes, and plans on how to achieve accessibility and dissemination of research findings should underpin the research from the start. While impact is often referred to as public and patient benefit, other forms of impact are also valid. Shorter term impact can include increasing knowledge or awareness, or changes in attitudes and motivation, which can contribute to changes in individual practice. Longer-term impact include the changes in policy, behaviour or practice that benefit patients and the public, which can take years [15].

## 1.1 The Wales COVID-19 Evidence Centre

Several new entities were set up to conduct rapid reviews of research evidence to aid decision-makers during the COVID-19 pandemic. Examples include the UK Health Security Agency COVID-19 Rapid Evidence Service, the COVID-19 Evidence Network to support Decision-making (COVID-END Global), and the WCEC [16–18]. With the remit of '*Good questions, answered quickly*', the WCEC was funded by Welsh Government and rapidly set-up (March 2021 –March 2023). The purpose of the WCEC was to collect research evidence and ensure this was accessible and rapidly available to the people making decisions for health and social care policy and practice in Wales during the pandemic, and in the later move towards recovery. The WCEC funding covered the WCEC core team, its partner research groups (researchers that conducted the evidence reviews) and public partner members [19], but not stakeholder time or commitment which was given *gratis* throughout the work.

Knowledge mobilisation and impact activities underpinned all the WCEC processes, which including stakeholder identification, research question prioritisation, rapid evidence reviews, and also rapid primary research and public involvement (the two latter are not included in this paper).

In this paper we have defined collaboration as an interdisciplinary cross-sector partnership that involves working together and coordinating between researchers, and stakeholders from institutions and or organisations, (e.g. WCEC and Welsh Government), which brings distinct and essential expertise to a project. We have defined co-production as working together with our stakeholders throughout a project (e.g. a rapid review), in processes including attending meetings, refining the research questions, identifying outcomes, discussion and interpretation

of the findings, interpreting policy and practice implications, reviewing and commentating on all outputs, (e.g. reports, infographics), and involvement in knowledge mobilisation processes.

## 1.2 Aims

The aims of this paper are to describe, evaluate and reflect on the processes for knowledge mobilisation and evidencing impact from the WCEC rapid research evidence reviews, describe lessons learnt, and make recommendations for best practice. This should serve as a resource for future crisis situations where similar Centres may be set up, and to inform future efforts enabling the use of research evidence by health and social care decision makers.

## 2.0 Materials and methods

Knowledge mobilisation and evidencing impact were a priority for the WCEC, with two members of staff employed to support this, and a member of the Welsh Government Technical Advisory Cell (TAC) working closely with the WCEC core team to facilitate engagement and communication between Welsh Government teams and the Centre [19].

Knowledge mobilisation processes were iterative, tailored to meet the requirements of stakeholders and included the following activities: 1) co-production and engagement with stakeholders, 2) stakeholder informed knowledge mobilisation, 3) wider dissemination, 4) associated knowledge mobilisation activities, and 5) tracking and evidencing impact (Fig 1). We describe each step below.

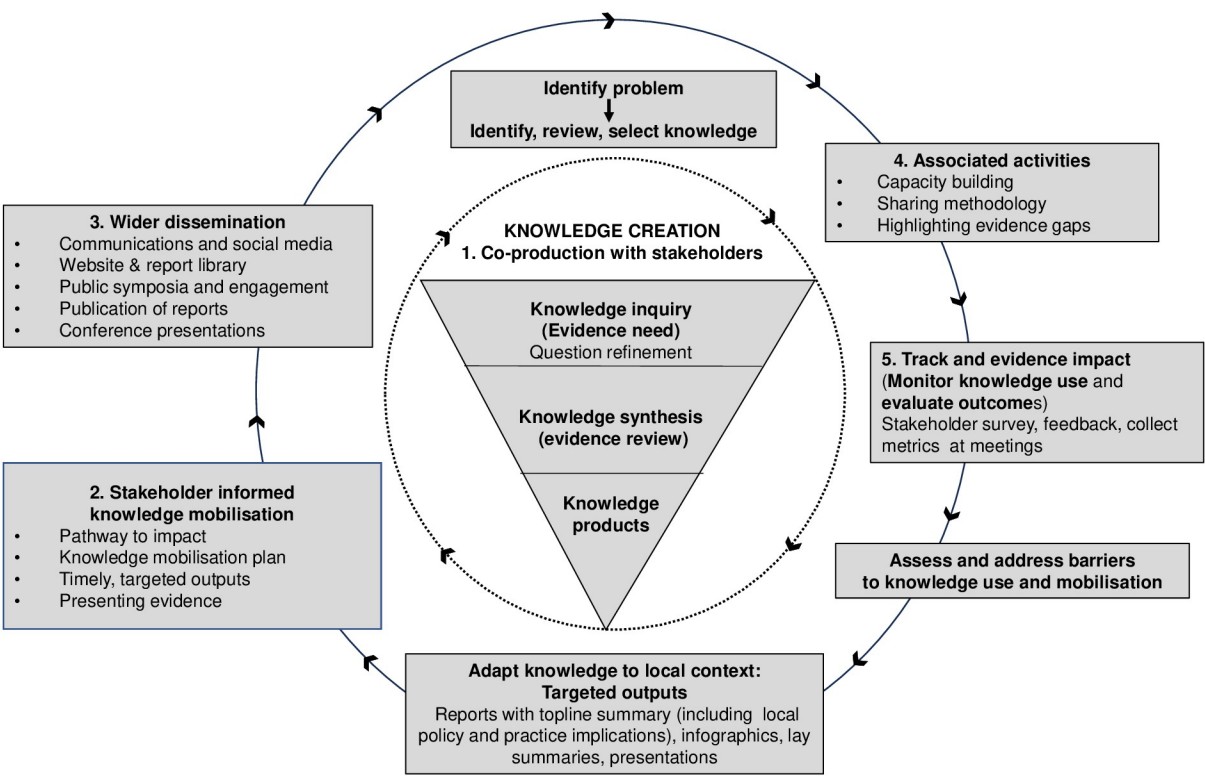

**Fig 1. Knowledge mobilisation and impact framework: The 5 steps and areas of activity.** Figure based on revised Knowledge-to-Action framework developed in 2006 by Ian Graham and colleagues.

## 2.1 Co-production and engagement with stakeholders (step 1)

The WCEC identified and reached out to stakeholder groups in Wales to identify research questions of the highest priority for health and social care decision-making during the pandemic and subsequent recovery period [19–21]. Stakeholders engaging with the Centre included teams from Welsh Government's TAC and Technical Advisory Group (TAG) (providing a role for Wales akin to "UK Government Scientific Advisory Group for Emergencies (SAGE)") described in their terms of reference [22], which provided coordination of scientific and technical advice to support Welsh Government decision-makers during the pandemic. Potential stakeholders within Welsh Government were identified via a stakeholder mapping exercise [19]. The TAC also had a boundary spanning role to promote communication between the evidence centre and policy-makers.

Other stakeholders included TAG sub-groups (e.g., Risk Communication and Behavioural Insights, Policy Modelling, and Environment) [19], Social Care Wales, NHS and social care leaders, public [21] and professional societies (Table 1).

Stakeholders' questions were prioritised, with those most urgent and likely to have an impact (e.g., inform health and social care policy or practice decisions) being taken onto the WCEC work programme (Fig 2) [20]. The WCEC contacted stakeholders to invite submission of new questions every 3 months. Additionally, the prioritisation process was flexible and responsive to accommodating new urgent questions outside the 3-monthly invite. The WCEC and stakeholders worked collaboratively to develop focused research questions and agree on outcomes, discuss evidence review findings and relevance to the local setting (Wales), identify policy and practice implications, write and review the final reports, plan and support knowledge mobilisation, and provide evidence of impact. During the review process, 1–3 representative stakeholders from each stakeholder group (that had a question accepted onto the WCEC work programme) were invited and attended a minimum of three on-line meetings with the WCEC. The meetings included an initial meeting to ensure the question was focused and that the outcomes were relevant to the stakeholders needs, a second meeting (one week later) to present initial findings and discuss the next review steps, and a third meeting to present findings and discuss knowledge mobilisation and impact [23]. If a key stakeholder was unable to attend a meeting, they were invited to send a representative from their group to attend on their behalf. These stakeholder meetings were a part of every review undertaken by the WCEC.

In addition to e-mail communication, co-production was enabled through the on-line meetings with key stakeholders during the review process, and involving stakeholders in dissemination of findings (e.g., Welsh Government evidence briefing sessions, and public symposia). The meetings with stakeholders and evidence briefing sessions included discussions about knowledge mobilisation and impact. Feedback to improve WCEC processes was also sought from stakeholders at the meetings, via e-mails and through a stakeholder survey.

## 2.2 Stakeholder informed knowledge mobilisation (step 2)

**2.2.1 Pathway to impact.** Stakeholders submitting questions to the WCEC were asked to identify the potential impact that answering a question would have (e.g., used to inform a particular policy or decision) and the timeframe that the evidence was needed by) [20].

**2.2.2 Knowledge mobilisation plans.** Final evidence reports were accompanied by a knowledge mobilisation plan developed with stakeholder input (Fig 3). The plans included 'stakeholder mapping' to identify key individuals involved in policy or practice decision-making, and further groups with an interest in the review topic. The plan identified relevant resources and dissemination methods which would be jointly supported by the WCEC team

**Table 1. Stakeholders engaging with the Wales COVID-19 Evidence Centre.**

| |
|---|
| **Welsh Government (WG) and Senedd (Welsh Parliament)** |
| TAC and TAG. TAG subgroups include the All-Wales Modelling Forum, Policy Modelling, Research and Development, Socioeconomic Harms, International Intelligence, Virology and Testing, Children and Young People, Risk Communication and Behavioural Insights, Environmental Science. <br> (Membership of TAC/TAG included Welsh Government, Public Health Wales, NHS Wales and academia; experts from public health, health protection, medicine, epidemiology, modelling, technology, data science, statistics, environment, microbiology, molecular biology, immunology, genomics, risk communication and behavioural insights, physical sciences, research). |
| Other WG teams: Long COVID Task Force, Equality and Human Rights Division, Homelessness prevention, Education (including Early Childhood Education and Care and Early Years Workforce, Childcare, Play and Early Years Division, Education and Public Services Group), Rural affairs team/Rural development division, Cost of Living Expert Group, Energy and climate change group, Planned Care Improvement and Recovery, Knowledge and Analytical Services, and the Health and Social Services group more generally. |
| Senedd (Welsh Parliament) Health and Social Care Select Committee, and Senedd Cross-party Group on Long-COVID. |
| **Social Care Wales** |
| Social Care Wales's Improvement and Development team |
| Association of Directors of Social Services (Wales) |
| **National Health Service (NHS) and other organisations aligned to health** |
| Vaccine equity committee (Welsh Government and Public Health Wales) |
| Public Health Wales Virology reference laboratory |
| Public Health Wales Vaccine Preventable Disease Programme team |
| Wales Health Board Medical Directors |
| All Wales Medicines Strategy Group |
| Academy of Medical Royal Colleges Wales, Royal College of Surgeons, Royal College of General Practitioners, Royal College of Podiatrists |
| Wales Cancer Network |
| National Health Service (NHS) leads in Wales including for primary care and cancer |
| General Medical Council-UK |
| Welsh Ambulance Trust |
| NHS Wales Shared Services partnership (Engineering team), and Infection control |
| Health Education and Improvement Wales, Deans of Medical Education (Wales) |
| UK Health Security Agency |
| NHS Wales: Directors of planning and finance, Diagnostics Board |
| **Members of the public and representative groups from under-served communities in Wales** |
| Service Users for Primary and Emergency Care Research Group |
| WCEC Public Partnership Group (8 members) |
| Members of the public attending WCEC public symposium (included workshops to identify research priorities). |
| Young people (from THE Centre for Development, Evaluation, Complexity and Implementation in Public Health Improvement (DECIPHER) ALPHA group (a research advisory group of young people aged 14 to 25 in Wales) |
| Disability Wales |
| Ethnic minority and Youth Support Team (EYST) Wales |
| Housing association (Taff Housing) |

and the stakeholders e.g., Welsh Government 'evidence briefing' sessions and themed public-facing symposia.

**2.2.3 Timely targeted outputs.** *Reports*. Reviews were conducted rapidly as required by stakeholders (up to 3 months for rapid reviews, and 1 week for rapid evidence summaries) [23]. The outputs of evidence reviews were presented in reports. Report templates were developed to ensure relevant information could be easily accessed by stakeholders. Reports were full scientific evidence review reports, but to aid access to key information by stakeholders, every

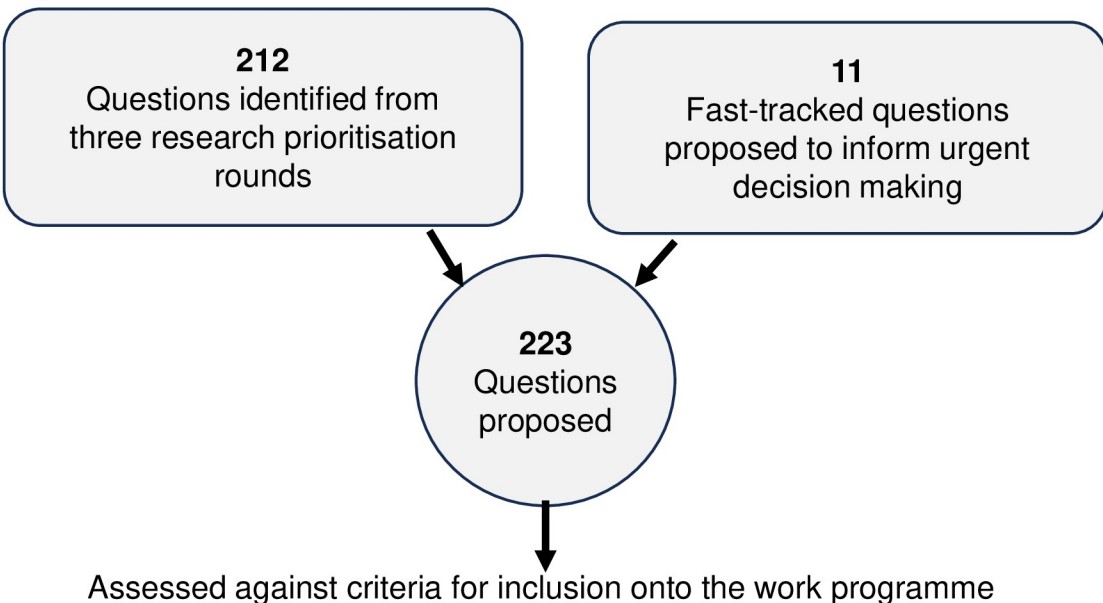

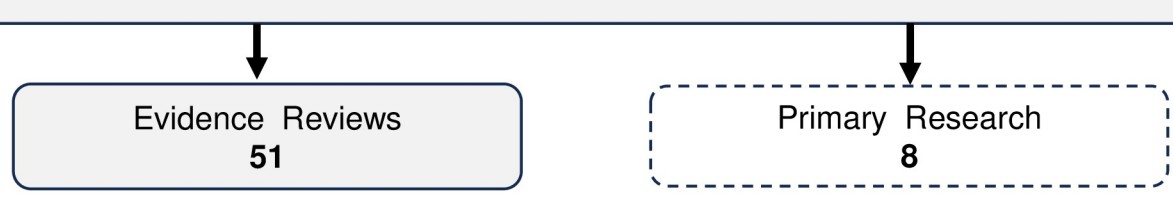

**Fig 2. Questions received, prioritised and accepted onto the centre work programme.**

report had 3 specific front pages. These included a short information box of review details, followed by a 2-page 'Topline summary' containing key information in an accessible language (Table 2). Stakeholders who attended the meetings during the review process, contributed and reviewed the Topline summaries to ensure the 2 pages contained the key information they needed, that the policy and practice implications were accurately reflected, and that the findings were accessible.

Stakeholders, including the WCEC public partnership group members, contributed to report reviewing and identifying the policy and practice implications. Reports were made available to stakeholders for use within their teams at the point of completion (pre-publication) but were watermarked to limit circulation beyond those teams until publication on preprint servers and in the WCEC library. Feedback on report structure, timeliness and accessibility of the information was collected via the stakeholder survey.

*Infographics.* When identified by stakeholders as being helpful for accessibility and dissemination purposes, a one-page infographic was produced by LD/FC using Prezi and Affinity

| Stakeholder mapping | Knowledge mobilisation plan | Resources | Timeline | Potential impact | Track Progress | Early Impact |
|---|---|---|---|---|---|---|
| Key stakeholders (e.g., scientific advisors, decision makers)<br><br>(Names and contact details) | Co-production with stakeholders:<br>• Timely report with identified policy and practice implications<br>• Presentation of findings at Evidence briefing<br>• Including in Advice to Ministers | • Watermarked report (pre-publication)<br>• Published Report<br>• Infographic<br>• Presentations of report findings | Aligned to stakeholder need e.g., Report available for internal use in 2 months | • Informed scientific advisers<br>• Change to policy or practice<br>• Longer term: Benefit to patients and communities | • Stakeholder feedback<br>• Survey<br>• Findings used and referenced in ministerial advice | • Informed policy change or plan<br>• Longer term: patient and public benefit<br>• Built trust and collaboration |
| Other groups that may have an interest in this area (e.g., Social care, Healthcare educators)<br><br>(Names and contact details) | Co-production with stakeholders:<br>• Share report and list of other relevant reports e.g., for cancer, social care<br>• Highlight research gap to research funders e.g., Health and Care Research Wales | • Published report<br>• Lay summaries<br>• Infographics<br>• Public symposia | • Report sharing: 1-2 months<br>• Symposia: 6 months | • Informed COVID related plans<br>• New knowledge<br>• New collaboration | Feedback from groups and key people via e-mail or at meetings | • Influenced change and increased knowledge<br>• Influenced funding call |
| Wider dissemination and communication | Co-production with Health and Care Research Wales and review team communication systems<br>• Social media<br>• Report published on pre-print server, and WCEC library<br>• Public symposia<br>• Newsletters | • Published report<br>• Lay summaries<br>• Infographics<br>• News story<br>• Blog | 1 week to 3 months | • Increased public knowledge, and awareness of WCEC and its work<br>• Opportunity for members of public to get involved in WCEC work | • Metrics: number of Tweets, downloads of reports, visits to website, stories by National media<br>• Number of participants and feedback | • Increased public engagement with WCEC<br>• Avoided duplication of work |

**Fig 3. Example of a knowledge mobilisation plan.**

Software. The infographics highlighted the main report findings, and the policy and practice implications in a single page, using accessible text and illustrations.

**2.2.4 Presenting evidence.** Review findings were shared via presentations to key stakeholders at the final stakeholder meetings, to the Welsh Government TAC and TAG, and at fortnightly on-line Welsh Government evidence briefing sessions. Evidence briefings were invitation-only events where questions could be posed to a panel (key stakeholders, WCEC public partnership group members, review team), and results and knowledge mobilisation discussed.

## 2.3 Wider dissemination (step 3)

**2.3.1 Communications and social media.** Health and Care Research Wales (HCRW) developed a communications strategy (including media and social media management),

**Table 2. Information contained within the front pages of the centre reports.**

| |
|---|
| **REVIEW DETAILS (Page 1)**<br>**Title of Report:**<br>**Report number and month/year completed:**<br>**Names of those involved in the report:**<br>**How to cite the report:**<br>**Disclaimer:** |
| **TOPLINE SUMMARY (Pages 2–3)**<br>**What is a. Rapid Evidence Summary/Rapid Evidence Map/Rapid Review:**<br>**Who is the report for:**<br>**Key Findings:**<br>**Quality of the Evidence:**<br>**Policy and/or Practice Implications:**<br>**Implications for Research:**<br>**Strength of the Evidence:** |

provided support for the WCEC website and events, and worked to feature WCEC in the media. Communication outputs for the reports including blogs, news stories, social media and event webpages were delivered by HCRW, raising the profile of WCEC and its publications across HCRW channels. A WCEC hashtag (#WalesCovidEvidence) was used to raise social media profile. WCEC's Public Health Wales and Health Technology Wales partner groups (NHS based) also undertook social media to raise awareness of WCEC outputs within their networks and websites.

**2.3.2 Website and report library.**   Health and Care Research Wales created, updated, and maintained an online WCEC section on their website in English and Welsh. The WCEC web-page hosted information about its work programme, methods, events, and links to news stories and WCEC newsletters.

Public facing documents including infographics, lay summaries, report Topline summaries, symposium presentations, newsletters, news stories, blogs and the WCEC work programme were translated into Welsh and hosted on the Welsh language version of the website. Full translation of reports into Welsh was available on request.

The website included an easily searchable (by theme, key words), bespoke report library. The library landing page for each report was a lay summary (written by members of the WCEC public partnership group) with a link to the full report and infographics, which could be downloaded.

A link to the WCEC website was included in the websites of other entities undertaking COVID-19 related rapid evidence reviews to raise awareness of the Centre's work and avoid duplication. This included the UK Health Security Agency Rapid Evidence Service and COVID-END.

**2.3.3 Public symposia and engagement.**   WCEC held three themed public facing, on-line 'Evidence into Practice Symposia' focusing on specific areas of its work including the effects of the pandemic on education, inequalities, and public involvement. Senior Welsh Government ministers or senior policy officials or science advisors opened each of the events. The Symposia programmes included question-and-answer sessions with an expert panel (Welsh Government scientific advisors, stakeholders, and members of the evidence review teams). At one Symposium, breakout groups were included to identify questions that were important to participants, with subsequent voting to rank the top 10 priority questions for inclusion in the WCEC question prioritisation process.

WCEC took part in two public engagement events and a series of focus groups to raise awareness of the Centre and to engage and involve under-represented groups in Wales in its work. Set up, promotion and conduct of these events was supported by HCRW members. (The public engagement is not included in this paper).

**2.3.4 Publication of reports.**   Initially, completed evidence review reports were published only in the WCEC library. From May 2022, reports were published on pre-print servers including MedRxiv (health and social care relevant reports), EdArXiv (education relevant reports), and Research Square (an environment relevant report). The WCEC library linked to the reports on the pre-print servers.

**2.3.5 Conference presentations.**   Members of the WCEC were encouraged to present methods and review findings at conferences to increase visibility of the Centre, its expertise and work to academic and public health audiences.

## 2.4 Associated activities (step 4)

Associated knowledge mobilisation activities included capacity building, sharing methodologies, and identifying evidence gaps. Evidence gaps and need for further research identified

during the review processes were clearly indicated within each report, and collected in a document, with an aim to share with research funders (e.g., Health and Care Research Wales and the National Institute of Health Research). COVID-19 related systematic reviews and preparation of infographics were offered as projects to medical students at Cardiff University. As the timeline of student projects would not have met the rapid turnaround required for many reviews, the questions they answered were less urgent. Stakeholders still received timely findings via a report and presentation, and students subsequently submitted the reports to the University and to peer reviewed journals.

## 2.5 Track and evidence impact (step 5)

**2.5.1 Stakeholder survey.** A stakeholder survey was developed to capture information and feedback including on WCEC processes, report structure, ease of information access, further suggestions for knowledge mobilisation, and using the review evidence. Participants were asked to select from a 5-point Likert scale for questions, and the data was analysed to calculate the frequency percentage. The survey also included open-ended questions to collect stakeholder feedback. The open-ended questions were optional and information was used to gather information for knowledge mobilisation plans and as example quotations (anonymised). No qualitative analysis was planned or conducted. All stakeholders (n = 44) taking part in the three stakeholder meetings during the review process were asked to complete the survey. Stakeholders were asked to completed the survey via e-mail following the report publication, and up to three e-mail reminders were sent.

Where stakeholders were involved in more than one evidence review, they were invited to include more than one report in their survey completion. The survey was open from the 2[nd] December 2021 to the 31[st] March 2023. No ethical approval was required for this survey as it was a service evaluation of the WCEC processes and outputs, and not conducted for research purposes.

**2.5.2 Collecting additional stakeholder feedback, and metrics.** The WCEC received a total of 223 questions from stakeholders (Fig 2). These included 11 urgent Welsh Government questions. Only questions (n = 58) with a clear pathway-to-impact (e.g., answering a question would help to inform Welsh Government advisors, a policy or plan) were accepted onto the WCEC work programme (i.e. research that would be conducted by the Centre) [19]. Where stakeholder surveys were not completed, information pertaining to impact and use of the report was collected via e-mail requests to the stakeholders involved in the stakeholder meetings, and during discussions at the on-line stakeholder meetings and the evidence briefing sessions, which were both attended by stakeholders. Where further information was needed and stakeholders did not respond to e-mails request or complete the survey, the boundary-spanning Welsh Government team member contacted Welsh Government stakeholders directly to obtain this information.

Reference of reports in stakeholder publications (e.g., Welsh Government TAC/TAG publications) was identified and collected. Identifying reference to WCEC reports in Welsh Government publications was enabled by having a boundary-spanning Welsh Government team member to help identify and search for these. Members of the WCEC core team also searched reports published on the TAC/TAG pages of the Welsh Government website for inclusion of WCEC reports.

## 3.0 Results

Note that quotes presented in the results section are taken from the stakeholder survey free text feedback, and are for illustrative purposes only.

### 3.1 Co-production and engagement with stakeholders (step 1)

In two years of operation (March 2021–23), the WCEC engaged and collaborated with 44 stakeholder groups and produced 51 reports from its rapid review work [19].

No stakeholders declined to participate in WCEC's work owing to lack of direct funding throughout its 2021–23 programme. There was good engagement from all stakeholders involved in the WCEC reviews. We defined good engagement as stakeholders or stakeholder representatives i) attending stakeholder meetings to help refine the research question [usually 'PICO' format [24]]) and agreeing relevant research outcomes, ii) agreeing deadlines for the evidence outputs, iii) interpreting review findings, identifying policy and practice implications, iv) providing input to knowledge mobilisation plans, v) supporting dissemination (e.g., sharing reports with colleagues) and vi) providing information on how the research evidence was used. Where requested, stakeholders or stakeholder representatives also participated in the additional dissemination activities (evidence briefings and public symposia). Poor engagement would have been defined by failure to attend (personally or via sending a representatives) the stakeholder meetings during the review process without which the review would not have been taken forward, and providing no contribution towards knowledge mobilisation or evidence of how the review findings were used.

Results from the stakeholder survey (n = 21) indicated that 19 (90.5%) stakeholders were 'very satisfied' with the engagement process and meetings, and two (9.5%) were 'satisfied'. (Details on survey sample and responses can be found below in section 3.5). Stakeholders reported value in being part of the review process, committed their time to attend the meetings and understood and trusted the review methods and findings e.g.:

> '. . .the WCEC representatives involved with the project were highly responsive. Their suggestions were constructive and they worked proactively to make meaningful progress that allowed our Welsh Government project team to quickly and conveniently locate relevant evidence that helped shape our policy proposals'

(S13).

Whilst positive towards the general approach, feedback suggested refining down-stream question prioritisation processes, so that the evidence review processes could be streamlined:

> '. . ..I think that . . .. there is no perfect solution or approach. There is an evident need to manage demand and expectation at every stage of the process if the WCEC is not to be over-whelmed. . . . .. there is.. a need to be more rigorous (even brutal) in requiring that questions are specific and narrow (and thus deliverable) from an earlier stage but also understand that this is not easy'

(S10).

### 3.2 Stakeholder informed knowledge mobilisation (step 2)

**3.2.1 Knowledge mobilisation plans.** Gathering information to support a knowledge mobilisation plan for each review worked well during discussions at stakeholder meetings, evidence briefings and via e-mail follow-up. For all the reviews, stakeholders or their representatives provided suggestions about which organisations and persons would be interested in the findings, and shared the reports with their groups.

Additional information for the knowledge mobilisaton plan was also gathered from the stakeholder survey when participants provided this information here.

Stakeholders also shared the reports within their teams and in some cases supported knowledge mobilisation more widely. For example, stakeholders from Social Care Wales arranged for findings from a rapid review of 'Innovations help to attract, recruit and retain social care workers within the UK context' to be presented at the 2022 Association of Directors of Social Services (ADSS Cymru) summer seminar and at a meeting of the ADSS Cymru Workforce Leadership Group. They also shared the report with the Social Care Wales workforce task and finish group (community capacity building), and with regional workforce boards (Directors of Social Services, workforce leads, managers and staff in Wales).

A further example is for a review conducted on 'the effectiveness of community diagnostic centers. Stakeholders including the Welsh Government lead for planned care improvement and recovery enabled the findings to be presented to relevant decision-making groups including at a TAG meeting and at a Welsh Government Health Executive meeting (NHS (Health Boards) Directors of Planning, and Diagnostics leads).

**3.2.2 Outputs and presentations.** The results from the stakeholder informed knowledge mobilisation activities are presented in Table 3.

## 3.3 Wider dissemination activities (Step 3)

The results of wider dissemination activities are shown in Table 4.

## 3.4 Associated activities (step 4)

Examples of Associated activities carried out by the WCEC are provided in Table 5. Four Cardiff University undergraduate medical students conducted COVID-19 systematic reviews, which have been submitted for peer-reviewed publication. 3 other students completed infographics to support WCEC evidence review reports.

Evidence gaps and need for further research that were identified in the review process are summarised in the topline summaries for each report and being made available on the website as a resource to researchers and research funders.

The WCEC shared its methodology and work programme to raise awareness of the Centre's work and avoid duplication. The September 2021 work programme was also referenced in a Welsh Government TAC summary of advice [31].

## 3.5 Track and evidence impact (step 5)

**3.5.1 Stakeholder survey.** 22 of 51 reviews were evaluated via the survey, and the remainder were evaluated through data collection following e-mail request, feedback at stakeholder meetings, and feedback provided by stakeholders to the WG liaison member of the core team.

All stakeholders (n = 44) who participated in the reviews and stakeholder meetings (1–3 per evidence review) were invited to complete the survey. However, stakeholders were variably and overall less engaged in the survey processes. Twenty-one (48%) stakeholders (mainly from Welsh Government) completed the survey (to December 2022) and two of these completed the survey for more than one evidence review. The participants included representatives from Welsh Government (policy lead and scientific advisors), Public Health Wales, the NHS and social care. Not all stakeholders answered every question. Two respondents dropped off towards the end of the survey. Nevertheless, some useful information was obtained regarding impact, and overall satisfaction with Centre processes including those aligned to knowledge mobilisation was high (Table 6), with feedback including:

*'I have seen excellent teamwork and a strong professional ethic that holds itself to account to provide the highest quality research and analysis'. . .. 'The model for delivery is one that is agile and flexible as well as speedy and there is a demand for this—albeit with an understanding of the caveats that come along with it'*

(S6).

There were no overtly critical responses, and a general consensus that the approaches used worked well as evidenced in Table 6.

**3.5.2 Collecting additional stakeholder feedback and metrics.** In some cases, it was necessary for the Welsh Government liaison person to gather further evidence of how the research evidence was used (if not included in a Welsh Government publication or testimonial received via e-mail or survey) e.g. if a key stakeholder had left or changed department. This was enabled as the liaison person was able to identify and contact relevant people within Welsh Government, which the WCEC was unable to do in some cases.

Table 3. Results from the stakeholder informed knowledge mobilisation activities.

| Activity | Results and feedback |
|---|---|
| **Reviews conducted and reports** | From April 2021 to March 2023, the WCEC conducted 51 evidence reviews and published 51 reports (in the WCEC library and on pre-print servers), which met stakeholders timelines (Fig 4). (The 51 evidence reviews included 10 rapid evidence summaries and 6 rapid evidence maps (delivered within 2 months, and 35 rapid reviews delivered within 3 months). Where very urgent, e.g. a review about the safety of ozone machines in schools, an evidence summary was delivered in 1 week). From 21 survey responses about the report structure and accessibility, 100% found the 'Topline summary' the most useful part of the report. The 'strength of evidence' section was deemed useful by over 90% of respondents, with several commenting on the importance of this area. *The highlighting of the strength of the evidence was excellent and particularly helpful. We often get evidence presented as gospel, it was good to see a more honest approach* (S18). Respondents suggested improving the reports for language accessibility and highlighting of key points: *'. . . Within full report the key findings potentially could be more clearly presented. If bold highlighting key text, then useful to have key words rather than whole bulleted statements for example'* (S12). |
| **Infographics** | 14 infographics were produced (example in Fig 5). Infographics were deemed a useful addition to support wider accessibility. Feedback for an infographic on the 'Effectiveness of community diagnostic centres' included: *'I think it's a really useful infographic for NHS Wales . . .. At a time when the system is so busy that Executives in particular may not initially have time to read the full evidence, the key points . . ..will certainly spark their interest'*(E1). An example of infographic use: The review findings and infographic for 'the evidence of direct harm from COVID-19 infection and COVID-19 vaccine in pregnant/post-partum women and the unborn child' were presented at a meeting of the midwives and maternity service leads in Wales. The infographic was deemed useful to support health professionals in their discussions with pregnant women regarding COVID-19 vaccination, and the group provided feedback to improve the readability. The infographic and report were promoted to midwives, to discuss with women, via the Public Health Wales COVID-19 immunisation programme page. |
| **Presention of review findings** | The results of 51 evidence reviews were presented to the key stakeholders at final stakeholder meetings. *24 Welsh Goverment evidence briefing sessions were held (October 2021 –January 2023), with an average of 33 participants. Feedback for the evidence briefings was positive: *'Both the evidence briefings and symposium were very insightful events that provided useful opportunities to share learning and develop links for future collaboration'* (S18). |

* Twenty-four evidencve briefing sessions were held to present and discuss review findings. It should be noted that evidence briefings were only introduced to the Centres processes at a later stage.

### Evidence review outputs

Between March 2021 and March 2023, the Evidence Centre produced the following evidence review outputs

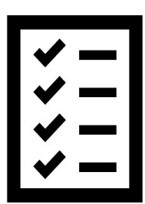

10
Rapid Evidence
Summaries

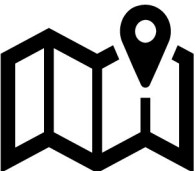

6
Rapid Evidence
Maps

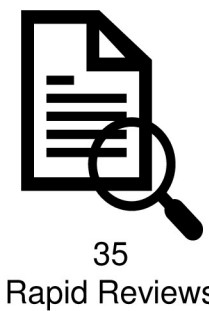

35
Rapid Reviews

**Fig 4. Evidence centre review outputs.**

The evidence reviews informed scientific advisors, health and social care policy and practice decision-makers, and their teams. Where the reviews identified a lack of published evidence, or that evidence was of low quality, the reports indicated the need for robust research and evaluation. In some cases, there was little evidence available in particular areas, and sometimes there was no further information other than that which was already known by stakeholders:

*Whilst the evidence of proven innovations was quite weak some of the evidence provided mirrored a lot of the initiatives that are already underway in Wales. The next phase is to further understand what practice is being adopted operationally and we have plans to do just that'*

(S14).

Evidencing actual impact from these types of reviews was challenging and often not possible.

Twenty-one review reports were referenced in Welsh Government advice and reports, and informed advisors and policy and practice decisions. The best impact examples of review findings being used to inform policy or practice decisions, are provided in Table 7.

## 4.0 Discussion

This paper describes the knowledge mobilisation processes of the WCEC, which were set up rapidly, and ensured that evidence was available to health and care decision-makers and advisers in Wales in response to the COVID-19 pandemic. Measures of success include that over 2 years, the WCEC worked closely with >30 stakeholder groups, including members of the public, to produce 51 evidence reviews, underpinned by knowledge mobilisation processes, which led to the evidence informing health and care policy and practice decisions during the pandemic. Below, we reflect on the five steps of our knowledge mobilisation framework, and follow this with lessons learnt by the WCEC.

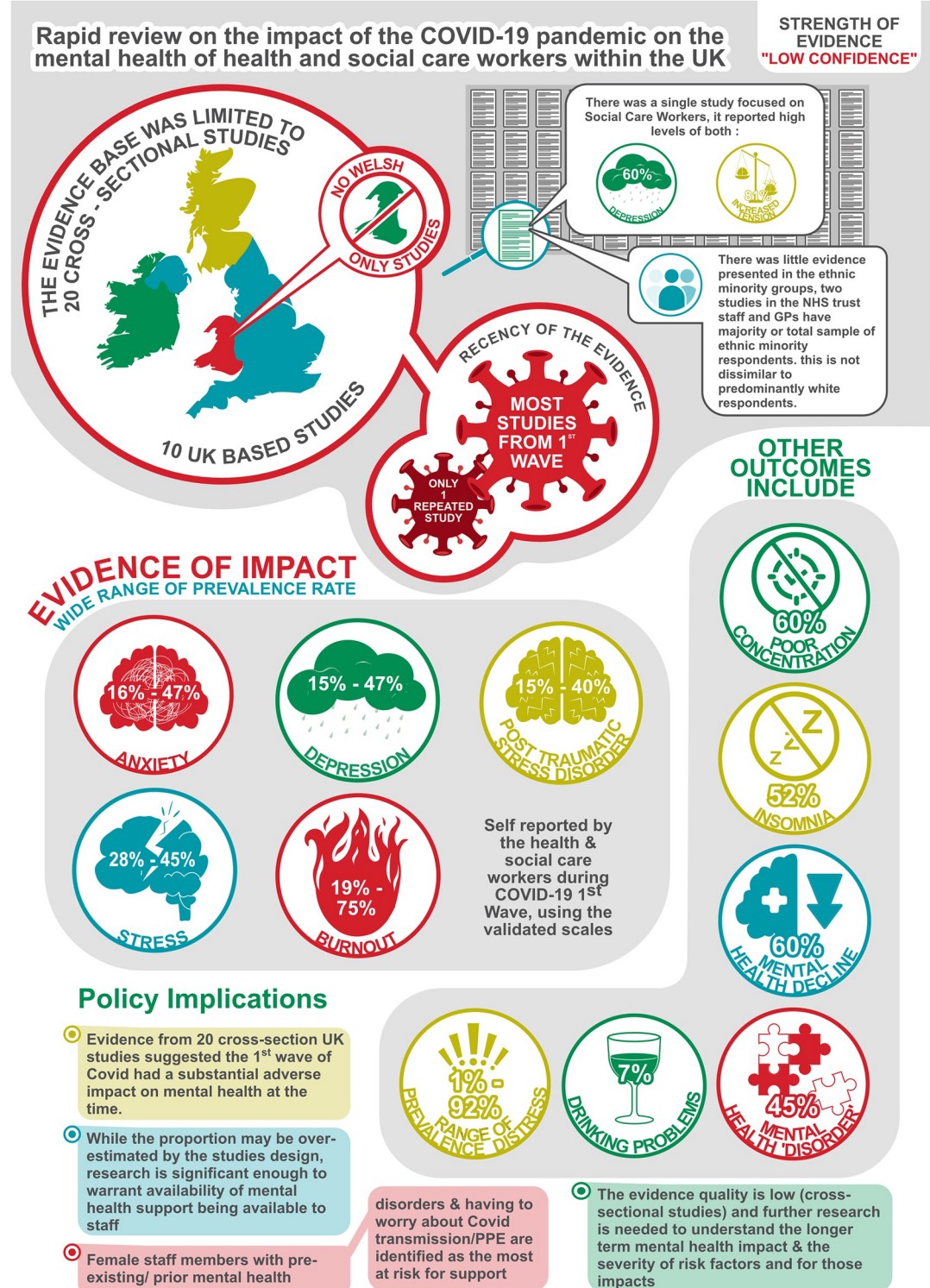

**Fig 5. Example of an infographic produced using affinity designer and publisher.**

**Table 4. Results of wider dissemination activities.**

| Method | Result |
|---|---|
| Communication and social media | • 21 reports have been included in Health and Care Research Wales news stories (to 20[th] Nov 2022).<br><br>• The Centre and its Director have featured on ITV Wales, BBC Wales News, BBC Wales Today and other local publications in Wales (Leader Live and Wrexham Live), potentially reaching audiences of 324,778 people.<br><br>• Health and Care Research Wales published 185 posts on Twitter in English and Welsh since March 2021. These posts have been engaged with 1487 times including likes, shares, and link clicks.<br><br>• A report on 'ozone disinfection machines in schools' was picked up by media [25–27]. |
| Website and report library | • The WCEC homepage was visited 7764 times by 5357 unique users (to May 2023)<br><br>• The report library, which was published online from March 2022, had 2006 views from 704 unique users (to May 2023) |
| Public symposia | 3 online Public Symposia were held and 70–80 people participated at each event. Stakeholders were involved as panel members for questions.<br><br>• **Evidence into Practice Symposium: The effect of the COVID-19 Pandemic on Education, Children and Young People**. December 2021<br><br>• **A Year of Impact** (This event included breakout groups to identify participants' COVID-19 priority questions). March 2022<br><br>• **Unequal impact, Fairer Recovery**. Findings from evidence reviews on the effect of the COVID-19 pandemic on groups including women and girls, lesbian, gay, bisexual, non-binary, intersex, asexual, aromantic, queer, and questioning (LGBTQ+) people, disabled people, prison and homeless populations, and racism in the NHS. September 2022 |
| Public engagement | WCEC had a stand (providing information about the Centre, the reports and the public partnership group) at 2 public engagement events organised by HCRW (268 and 532 attendees). These activities were supported by members of the WCEC Public Partnership Group. |
| Publication of reports | • Tracking the metrics of reports published on MedRxiv indicated that the reports were widely viewed and accessed: e.g. In the first week, the rapid review of 'What interventions or best practice are there to support people with Long COVID, or similar post-viral conditions or conditions characterised by fatigue, to return to normal activities', achieved 174 pdf downloads, had a blog in 'Medical Science News', and was included in a news roundup from the ME Association [28–30].<br><br>• 2 peer-reviewed papers were published from rapid evidence review work. |
| Presentations | The WCEC core team and partners presented about the Centre and findings of reviews 54 times to various groups, in addition to presentations at evidence briefing sessions and stakeholder meetings. |

## Step 1: Co-production and engagement with stakeholders

The successful engagement and co-production with a wide range of key health and social care stakeholders in Wales may not have happened outside the urgency and unique requirements of the pandemic. The pandemic directly led to the set-up of the new Welsh Government TAC and TAG, who the WCEC was able to work with closely. Having a liaison person who was embedded in both Welsh Government and the WCEC was crucial, and the benefit of having a person to span boundaries has been evidenced elsewhere [40]. The other health and care stakeholders also had an urgent need for evidence, which helped forge relationships and aided co-production and engagement.

The responsiveness and involvement of stakeholders and members of the public partnership group were key to focusing the research questions, which were often too broad to be answered in the available amount of time [23]. The expertise of the stakeholders and members of the public partnership group was also essential to contextualise the review

**Table 5. Examples of associated activities conducted by the WCEC.**

| Activity | Example |
|---|---|
| Student involvement in WCEC research work to gain experience of research and knowledge mobilisation | 4 Cardiff University (CU) medical students conducted COVID-19 related systematic reviews for the WCEC (pandemic impact on racism in the NHS, interventions to mitigate against racism in the NHS, pandemic impact on prison, and pandemic impact homeless populations. The students also presented findings at stakeholder meetings, at WG Evidence briefings and at public symposia. 3 CU students completed infographics to summarise the findings for three rapid reviews. |
| Identifying and sharing research gaps | All evidence reviews identified gaps where further research was needed. These gaps were highlighted in the Topline summary of each report. Gaps are being collated, and shared on the website as a resource for researchers and research funders. |
| Developing rapid review methodologies with research partner groups and sharing Centre methods. | Weekly meetings were held with research partners to develop the rapid review methodologies. All methods were shared e.g. with the UK health Security Agency, stakeholders and presented at Conferences. Our methods were described on the website, shared on request. |
| Sharing the questions on the WCEC work programme to raise awareness and avoid duplication | Shared on our website, and with the UK Health Security Agency Rapid Evidence Service, National Institute of Clinical Excellence (NICE) Wales, and COVID-END. |

findings, and support and expedite knowledge mobilisation. Involvement during the review process also helped understanding and built trust in the review results and the WCEC methods.

Moving beyond the pandemic, the need for evidence to address issues like the long health service waiting lists, is ongoing, though the time frames may not have the same immediate urgency. A key challenge will be the maintaining the stakeholder relationships to ensure future research will contribute towards evidence informed policy and practice. Already structures and teams set up during the pandemic have changed. For example, the TAC and TAG have now dissolved, though their strength and contributions were realised, and a new Welsh

**Table 6. Survey responses for questions related to knowledge mobilisation.**

| Question | Option 1 Number (%) | Option 2 Number (%) | Other options Number (%) |
|---|---|---|---|
| **Stakeholder engagement process** (n = 21) | Highly satisfied 19 (90.5%) | Satisfied 2 (9.5%) | |
| **Timeliness of the report and information** (n = 21) | Highly satisfied 18 (85.7%) | Satisfied 3 (8.3%) | |
| **I trusted the information in the report** (n = 22) | Strongly agree 20 (90.9%) | Agree 2 (9.1%) | |
| **The report presented required information in a clear manner** (n = 22) | Strongly agree 13 (59.1%) | Agree 8 (36.4%) | Neither agree nor disagree 1 (4.5%) |

The data from the Stakeholder survey are accessible on the data repository Figshare (Wales COVID-19 Evidence Centre 2021–2023 Stakeholder survey).

Table 7. Examples of WCEC rapid evidence reviews: Impact and pathway to impact.

| *Report | Evidence of impact |
|---|---|
| **Can we better quantify the relative risk of COVID-19 transmission in enclosed, semi-enclosed and outdoor environments: a rapid evidence summary.** May 2021 | • Included in the Welsh Government '**Technical Advisory Group COVID-19 Restriction Review, (Advice on the proposed relaxations considered as part of the 22 April 2021 review)'** [32]. |
| **Vaccination uptake (barriers/facilitators and interventions) in adults from under-served or hard-to-reach communities: A rapid evidence summary.** July 2021 | • The vaccine inequities report was referenced and informed '**vaccine passport advice'** from the TAC. (September 2021) [33].<br>• Findings shared with the Welsh Vaccine Equity committee including the Deputy Health Minister. |
| **What is the evidence of direct harm from COVID-19 infection and COVID-19 vaccine in pregnant/post-partum women and the unborn child: a rapid evidence summary** July 2021 | • Information (infographic) was used by midwives in their consultations with pregnant women.<br>• Referenced in the **Chief Medical Officer for Wales Coronavirus control plan: autumn and winter 2021 update** [34]**.**<br>• Referenced in a message to the Primary Care and Community Services Division, which includes all GP Practices in Gwent and all GPs on the ABUHB Performers List (November 2021).<br>Feedback from our stakeholders included, *'Findings bolstered and stimulated internal conversations on concerns around vaccine uptake in pregnancy and required actions. Also, facilitated action arising from the WG Winter plan via the Chief Medical Officers Coronavirus control plan: autumn and winter 2021 update'* |
| **The effectiveness of infection prevention and control measures applied in education and childcare settings for children: a summary and critical appraisal.** August 2021 | • Findings informed a **strategic framework and action in Education policy** aimed at operating schools and education settings safely during the COVID-19 pandemic. (October 2021).<br>• Findings informed shaping of a policy regarding a return to school for children in the autumn term after lockdown. |
| **The efficacy, effectiveness and safety of SARS-CoV-2 disinfection methods (including ozone machines) in educational settings for children and young people. A rapid evidence summary.** September 2021 | • Review findings formed a key part of the WG response to use of ozone disinfection machines in schools. Evidence was used to advise the Ministers for Health and Social Services and Education and influenced the WG decision to use carbon dioxide monitors in schools over ozone disinfection machines.<br>• The report is referenced in the Welsh Government 'Technical Advisory Group: evidence review of ozone generators including appropriateness as mitigation in classrooms' [35]. |
| **A rapid review of strategies to support learning and wellbeing among 16–19-year-old learners who have experienced significant gaps in their education as a result of the COVID-19 pandemic.** September 2021 | • Findings were used by the Welsh Government **COVID-19 Recovery group for Post-16 Education** to help compile a list of suggested strategies to be explored with stakeholders to support learner wellbeing and the progress of their learning in the wake of the pandemic. Stakeholders confirmed the report influenced a project plan and shaped the support measures they are developing in partnership with education and training providers.<br>• Evidence included in the Welsh Government '**Technical Advisory Cell: Summary of advice'** (November 2021), and '**Review and Renew and Reform: Post-16 and Transitions Plan'** (Section 7, Learning from the pandemic (March 2022) [36, 37]. |

*(Continued)*

**Table 7.** (Continued)

| *Report | Evidence of impact |
|---|---|
| **Barriers and facilitators to the uptake of personal protective behaviours in public settings: a rapid evidence summary.** January 2022 | • Referenced in 2 Welsh Government reports including: **'Updated advice from the Technical Advisory Group** (January 2022), **and Chief Scientific Advisor for Health on the evidence for the use of COVID Passes** (December 2021 [38]. |
| **What innovations can address inequalities experienced by women and girls due to the COVID-19 pandemic across the different areas of life/ domains: work, health, living standards, personal security, participation and education. Rapid review.** January 2022 | • The report was used in discussions with equality leads in Welsh Government who were working on a renewed **'Advancing Gender Equality in Wales Action Plan'** (2022).<br>• The findings informed the Welsh Government Gender equality subgroup and the plan to tackle gender inequalities in Wales. Feedback included, *'The report confirmed some of which is known already and where the gaps and priorities are which is good. Also highlighted further research and robust evaluation needs and gives the team some ideas about innovations to try out. There is a Advancing Gender Equalities in Wales Plan which is being renewed in light of COVID. This plan may include changes to reflect the identified innovations and current gaps. The report highlighted and confirmed the gaps which are currently not being addressed'* |
| **Innovations to help attract, recruit and retain social care workers within the UK context.** A rapid review. January 2022 | • Included in Welsh Government **'TAC: Summary of advice'** (January 2022) [39].<br>• The theme of this report led to this area being a main topic at the Association of Directors of Social Services (ADSS Cymru) summer seminar (2022), where the findings were presented and discussed.<br>• In February 2022, the report was shared with UK Parliament (Health and Social Care Committee, House of Commons) to inform their enquiry–'Workforce: recruitment, training and retention in health and social care'.<br>• The report is also informing further research by Social Care Wales: 'Feedback (March 2023), *'We've already made use of the work on recruitment to inform a bigger piece of research we're currently doing around recruitment and retention'* |
| **Impact of the COVID-19 pandemic on the health and access to health care of disabled people: a rapid evidence map and rapid review.** March 2022 | • Showed that the risk of death involving COVID-19 was three times greater for disabled people than non-disabled. Findings were used to **inform the** Welsh Government's **Disability Rights Taskforce**, and highlighted the need for more research into how services recover from the pandemic. |
| **The effect of vaccination on transmission of SARS-CoV-2 (COVID-19): a rapid review**. March 2022 | • Findings referenced in the updated advice from the **Technical Advisory Group and Chief Scientific Advisor for Health on the evidence for the use of COVID Passes** [38].<br>• As COVID-19 transmission and strains were evolving, it was decided it would be important to continue this work to continue to provide updated evidence. For this, the WCEC worked together with the UK Health Security Agency (UK HSA). |
| **What is the impact of the COVID-19 pandemic and restrictions on LGBTQ+ communities in the UK and what actions could help address these: rapid evidence map.** April 2022 | • The evidence summary was used to inform the **Welsh Government LGBTQ+ Action Plan for Wales.** |

(*Continued*)

Table 7. (Continued)

| *Report | Evidence of impact |
|---|---|
| **A rapid review of the effectiveness of innovations to support patients on elective surgical waiting lists.** April 2022 | • Informed the Welsh Government plan for tackling waiting list backlog '**Our programme for transforming and modernising planned care and reducing waiting lists in Wales'** April 2022<br>• Feedback from stakeholders includes', *'The work you have done has helped us think about the pre-hab aspect of this plan and the priority to change from a waiting list to a prep list. The work you have done will be used for the implementation and the development of the actual solutions to the plan's priorities'*<br>• Welsh Government stakeholders informed us that when developing their approach to 'waiting well', the report findings would be a great help as they seek to develop and embed this. |
| **Barriers and facilitators to cancer screening uptake in under-served populations: a rapid review.** June 2022 | • Requested by the Screening Division of Public Health Wales and Velindre NHS trust. A number of research question on cancer screening participation were also submitted during workshops at our public symposium.<br>• Public Health Wales stakeholders confirmed that the findings helped them to identify if they were missing anything in their current practice and was helpful for them to understand where the gaps were and where further work may be needed.<br>• Findings are also informing interventions and campaigns designed to encourage people to take part in cancer screening which may need to be adapted after the pandemic. |
| **A rapid review of what innovative workforce models have helped to rapidly grow capacity for community care to help older adults leave hospital.** August 2022 | • This work informed Welsh Government policy and planning in this area in the Autumn of 2022, including the '1000 Beds' plan.<br>• Social Care Wales shared the report and raised awareness with the Social Care Wales workforce task and finish group looking at community capacity building. |

*All reports are available in the Wales COVID-19 Evidence Centre library [18].

Government Science, Evidence and Advice division has been set up, which includes members of TAC and TAG.

Following the end of funding of the WCEC, we received 5 further years of funding from Welsh Government via Health and Care Research Wales, and are now funded as the 'Health and Care Research Wales Evidence Centre (2023–2028) [41]. This enables us to continue longer term close collaborations with the stakeholders we have worked with in the WCEC to enhance use of our report findings, and collect and evidence longer term evidence of impact. We also continue to work with public members and currently the new Evidence Centre has a public partnership group who are involved in all the Centre processes.

To improve engagement and communication with policy and practice decision makers going forward, the new Evidence centre has now implemented ongoing, additional short meetings with stakeholders, which focus on service evaluation, knowledge mobilisation planning and impact tracking. The new Centre also now has additional contact between the review team and stakeholders during the review process (e.g. to discuss the review protocol), and this helps to manage and clarify expectations.

## Step 2: Stakeholder informed knowledge mobilisation

Involving stakeholders in knowledge mobilisation processes was crucial to enable evidence dissemination, and track use of the evidence. It enabled much wider sharing of evidence and links to be built with groups outside academia, which may not have been so readily accessible otherwise.

Knowledge mobilisation becomes particularly challenging where organisations are large and diverse and requires domain-specific strategies. Working with stakeholders to disseminate evidence ensures that evidence is seen as coming from a trusted source and reaches appropriate audiences. For example, the WCEC worked together with Social Care Wales knowledge exchange and research teams to disseminate evidence across social care organisations.

Reports of the evidence review findings, included the two-page Topline summary of headline findings. This was directly targeted to address the needs of stakeholder and their feedback was positive. The Topline summary was based on other evidence policy briefs including the BRIDGE Evidence-informed framework for effective information-packaging to support policymaking [9]. Moving forward, it may be useful to include economic considerations, and qualitative research capturing public/patient experiences alongside results of evidence synthesis [1].

In addition to tailored and targeted messages, training may increase evidence uptake by health managers and policymakers [42].

## Step 3: Wider dissemination

Wider dissemination processes increased access to our reports and work programme, and potentially avoided duplication of work by other Centres reviewing pandemic related evidence. It is worth noting that the wider dissemination processes required a considerable amount of time (e.g., developing and maintaining website and library, editing and publishing of reports, social media, organising public events and evidence briefing). Further evaluation to understand the meaning of metrics, such as downloads and social media mentions would have been useful. It is recognised that while passive knowledge dissemination increases access to evidence, it may not have any effect on uptake [42]. Future planning of wider dissemination with robust outcome measures and evaluation is needed.

Holding breakout sessions within one of the online public symposia worked well to engage wider public and identify their questions and priorities; public participants engaged in the follow-up processes including ranking their top 10 questions, which were included in the WCEC question prioritisation process.

To make the work of the WCEC more widely accessible, we produced a public facing legacy report [43]. This includes information about the Centre structure, methods, reviews, knowledge mobilisation and impact in an accessible format. The lay summaries accompanying each review were written by members of the WCEC public partnership group, and together with the infographics, increase accessibility of the review findings.

A suggestion for wider dissemination of findings in a different format was made by a public member (e.g., review findings relevant to children and young people could be made into a poem). We acknowledge that wider dissemination methods such as creating stories, poems or videos would likely be valuable to make findings more accessible and engaging to wider groups. However, this was not possible at the time due to resource constraints (available staff time and additional funding). Videos of our methods relating to knowledge mobilisation, impact and rapid reviews are now available on our new Centre website [41].

## Step 4: Associated activities

Involving students in the work of the WCEC was valuable for both the WCEC and students. The students gained experience not only of conducting systematic reviews but also experience

of knowledge mobilisation processes and of working with and providing evidence that informed stakeholders. Both the reviews and infographics produced by students were a valuable addition to the Centre outputs, and the students presented their review findings at Welsh Government evidence briefing sessions. The more in-depth requirements for academic student projects may not suit evidence synthesis where evidence need is very urgent as projects typically take longer to complete.

Identifying further research needs and gaps in evidence reviews should be valuable for informing both researchers and research funders. The funding organisations have specific requirements for considering such further research and evidence priorities, which should be considered.

## Step 5. Track and evidence impact

The best evidence of impact was for questions that came from the Welsh Government TAC and TAG groups, where the evidence was required to inform a specific plan, programme or guidance, and where knowledge mobilisation and gathering impact evidence was supported by the WCEC liaison person and stakeholders from the outset. While impact could be evidenced to the point of informing health and social care decision-makers (a recognised impact outcome) [15], evidence of how the research benefited public and patients would require future tracking and data analysis, which was not achievable within the timeline of the WCEC (2021–23), but which will be ongoing as part of the new Evidence Centre work. The unique environment of the pandemic and the close collaboration and co-production helped outputs to be used to inform decisions. However, moving forward, gaining a better understanding of barriers and facilitators to evidence use, how policymakers process evidence, and how other factors influence their decisions, will be crucial [11, 14].

Closer 'knowledge brokering' and embedding people to increase cross-community interactions may increase utilisation of evidence and impact [40]. Trade-offs between the resources required to enable this and increase in impact would be interesting to evaluate.

## The framework for our knowledge mobilisation processes

The 'Knowledge to Action' (KTA) framework was used to inform our model and processes, as the stages aligned to many of our intended processes, and the model could be adapted to our rapid requirements (Fig1). The original KTA framework includes a knowledge creation process and a seven phase action cycle including 1) identifying the problem or issue to change, selecting knowledge to address the issue, determining the gap between knowledge and practice, 2) adapting the knowledge to your context, 3) assessing barriers and facilitators to knowledge use, 4) selecting an implementation strategy to make changes, 5) monitoring knowledge use, 6) evaluating outcomes and 7) sustaining the change or use of knowledge [6]. The time constraints and volume of evidence reports limited our implementation of some of these actions. However, our experiences and learning, in addition to reflection on other models of knowledge mobilisation and communication will allow us to evolve our processes as we move forward [42].

## Lessons learnt by the Evidence Centre

As demonstrated by the example of the WCEC, it is possible to rapidly set up a new evidence unit, which can provide timely evidence to policy and practice decision making. While this was facilitated by the urgency of the pandemic, the model and lessons learnt are valuable to inform future initiatives and good practice during crisis situations.

During the lifetime of the WCEC (2021–2023), we captured challenges, enablers, and lessons learnt in relation to our knowledge mobilisation and impact processes. Our reflections are provided in Table 8.

Table 8. Reflecting on the learning from the WCEC knowledge mobilisation and impact processes.

| Area | Enablers, challenges and reflection |
|---|---|
| **Engagement and co-production with stakeholders** | • Working together with stakeholders from the outset is essential if research evidence is going to be relevant, timely and used by stakeholders.<br>• Co-production was essential to clarify the timeframe, research question, identify priority outcomes and identify the policy and practice implications.<br>• The close working relationship with the Welsh Government's TAC and TAG ensured that the reviews were impactful e.g., directly informing Welsh Government policy officials in developing policy and plans.<br>• Close stakeholder involvement helped to build relationships and trust in the outputs of the reviews and the methods used.<br>• Ensuring stakeholders were involved during the evidence synthesis process (e.g., at stakeholder meetings) enabled an understanding of the methodology, its limitations and trust in the evidence outputs.<br>• Ensuring that stakeholders were able to attend and contribute to the stakeholder meetings could be challenging. This was mitigated by ensuring WCEC team flexibility and stakeholders sending representatives.<br>• Having a boundary spanning person, e.g., between the WCEC and Welsh Government, may be essential to enable good collaboration and co-production with Welsh Government groups, for aiding knowledge mobilisation, and for evidencing and tracking impact. |
| **Stakeholder informed knowledge mobilisation and accessibility of findings** | • Knowledge mobilisation processes should be flexible to meet stakeholder need. Knowledge mobilisation is a 2-way process and should underpin the work from start to finish—we can push the evidence out, but key stakeholders need to pull it into their practice. The experiences and knowledge of stakeholders and public representatives is invaluable.<br>• Stakeholder input into the knowledge mobilisation plans was crucial to ensure findings reached relevant groups. They also provided considerable input to identify other groups that may be interested in the reports as part of the wider dissemination processes.<br>• Welsh Government evidence briefing sessions were useful to raise wider awareness of the findings, and to discuss policy and practice implications, knowledge mobilisation and impact.<br>• Welsh Government evidence briefings were useful during the COVID-19 pandemic where stakeholders were meeting frequently as teams (such as TAC and TAG). However, moving out of the acute phase of the pandemic, it was more challenging to ensure attendance, and presenting at stakeholders' team meetings saw more attendees.<br>• Members of our public partnership group contributed equally as stakeholders and were able to fit in with the rapid timelines required. Their contribution was also invaluable to ensure the evidence was accessible to lay audiences. |
| **Ensuring accessibility of findings** | • Accessibility of findings is essential if stakeholders are to use the evidence. The key findings, implications and strength of the evidence were summarised in an accessible language within the 2-page Topline summary, which was reviewed by stakeholders and well received.<br>• Infographics were useful but the additional time required to produce these should be factored in. A clear need for the infographics should be evidenced.<br>• Even outputs made accessible to the general public (e.g., lay summaries, infographics, our on-line public events) may not be readily available to some people e.g., those who do not have access to electronic and digital media. Public-facing engagement events may mitigate this, and consideration of the additional time and resources needed to do this well should be planned.<br>• Other methods of increasing the accessibility of findings e.g., poetry, storytelling, preparing videos, producing booklets which could be picked up in pharmacies may be useful. Any approaches should be accompanied by robust evaluation to assess effectiveness. |
| **Potential impact and pathway to impact** | • To ensure that research findings feed into urgent decision-making processes, the timelines, outcomes and the potential impact and pathway to impact should be identified and agreed with the stakeholders from the outset.<br>• Meeting the timeframe in which the relevant research output is needed by stakeholders is essential for evidence to be used and impact realised.<br>• Tracking impact and use of information by Welsh Government advisors and decision-makers was occasionally challenging where the key stakeholders involved had moved to different departments or left. Having a Welsh Government member 'boundary spanning' between the WCEC and Welsh Government was invaluable to help identify new group leads and contact Welsh Government members to collect evidence of impact.<br>• Impact beyond the informing of a policy or plan i.e., actual patient and public impact can take years to be realised. Plans for long term follow up should be considered.<br>• Evidencing the impact of wider dissemination can be challenging. Asking people to enter details for further contact could help examine what numbers of downloads and visits to a site mean.<br>• Social media highlighted the existence of the Centre and its work; however, evaluation of evidence of impact from social media was not done. Tweets by Twitter users could be analysed for number, type of user, likes and dislikes to gauge reach and impact.<br>• Involvement and contribution of public members was possible within all the WCEC review and knowledge mobilisation processes despite the rapid nature of the work. |

(*Continued*)

**Table 8.** (Continued)

| Area | Enablers, challenges and reflection |
|---|---|
| **Publication of reports** | • During the pandemic a delay of even 3 days would often not have been acceptable. It was essential to provide stakeholders with a watermarked (confidential) report prior to final editing and publication. Publishing in the WCEC library was immediate and could assist referencing in Welsh Government reports.<br><br>• Initially reports were hosted on the WCEC website and made accessible to view and download. However, if publishing in this way, reports do not have an individual identifier number (doi) so may not be picked by a Google search or search of published evidence resources. Additionally, when trying to publish the review in a peer-reviewed journal the reports can be identified by a journal's plagiarism software, which precludes the peer-reviewed publication.<br><br>• Publishing on pre-print servers was relatively rapid (3–7 days) and many journals accept publications previously available in this way. The reports also received a unique doi number and the preprint servers collect metrics including views, downloads, reference in policy documents and social media references i.e., in Twitter.<br><br>• When producing rapid reviews, the requirements and criteria of peer reviewed journals and the research and impact requirements of academia should be considered. Our review methodology is aligned to recognised rapid review methodology and a number of the reviews have now been published in peer reviewed journals [23]. |

Lessons that will be taken forward include i) refining our engagement and communication with policy and practice decision makers, and ii) involving stakeholders in the development of the knowledge mobilisation plan from the beginning of the collaboration, developing robust and longer-term evaluation plan from the outset to gather insights and ensure relevance. We will also plan activities to better include and engage members of the public especially from under-served communities and those digitally excluded in Wales. Getting a public voice and stories of personal experiences would likely be a powerful addition to the evidence [7]. This should be feasible when not having to meet the quantity and urgency of work that needed to be undertaken during the public health emergency of the pandemic.

### Strengths and limitations

Many of the strengths and limitations are described for each of the knowledge mobilisation steps above and in Table 8.

Strengths included the close involvement and co-production with stakeholders, including public partnership members. This ensured that evidence outputs were targeted, timely and used.

The WCEC process evaluation survey to better understand use and impact of the evidence reviews is a limitation and while providing some insight, could not contribute to robust evaluation. Methods to better evaluate different dissemination methods to assess their impact and effectiveness should have been planned and implemented as part of the processes from the start.

A further limitation warranting wider debate is what is recognised as impact. While rapid reviews could count as impact for informing policy and practice, the impact requirements for academia need to be further considered. Academia necessitates the publication of research in peer reviewed journals for both career development, and to underpin impact cases included in the UK Research Excellence Framework (which evaluates research impact of British Higher Education Institutions). While it is possible to further develop the rapid reviews in line with requirements of peer reviewed journals, this was only possible for 2 reviews during the lifetime of the WCEC, primarily owing to time constraints.

## 5.0 Conclusion

The WCEC demonstrated that an Evidence Centre could be set up within a rapidly moving pandemic, with knowledge mobilisation processes ensuring that evidence reviews successfully informed health and social care decision-makers. Lessons learnt will be useful for informing future set-ups of such Centres during crisis situations.

The value of the WCEC work to inform evidence based decisions and practice has been recognised with a further five years of funding (from April 2023) as the 'Health and Care Research Wales Evidence Centre' [41].

## Acknowledgments

We would like to acknowledge the WCEC research partners from the Wales Centre For Evidence Based Care, the Specialist Unit for Research Evaluation, the Public Health Wales Evidence Service, Health Technology Wales and the Bangor Institute for Health and Medical Research, the BioComposites Centre, and the Cedar Health Technology Research Service for conducting the evidence reviews and presenting findings at stakeholder meetings including the Welsh Government evidence briefings and WCEC public symposia. We would like to thank our students Tomi Adewole and Kismet Lalli for their work, and also

Peter Bragge, Lauren Elston and Sophie Hughes for their help in designing the stakeholder survey.

We would also like to acknowledge the members of our Public Partnership Group (PPG) and Barbara Harrington and Julie Hepburn from the Wales Primary and Emergency Care (PRIME) Service Users (SUPER) group for writing the lay summaries. Also, to members of the PPG for attending meetings with stakeholders and evidence briefings, Deb Smith for supporting our public engagement event, and the Welsh Government TAG secretariat in arranging evidence briefings.

We would like to thank all our stakeholders for giving their time and expertise and participating in knowledge mobilisation.

Lastly, we would like to thank the Health and Care Research Wales communications team for their support of the WCEC.

## Author Contributions

**Conceptualization:** Micaela Gal, Alison Cooper, Natalie Joseph-Williams, Elizabeth Doe, Ruth Lewis, Rebecca Jane Law, Sally Anstey, Amy Walters, Robert Orford, Brendan Collins, Andrew Carson-Stevens, Adrian Edwards.

**Data curation:** Micaela Gal, Alison Cooper, Elizabeth Doe, Rebecca Jane Law, Amy Walters, Jane Greenwell, Ffion Coomber.

**Formal analysis:** Micaela Gal, Elizabeth Doe, Nathan Davies, Amy Walters, Lisa Trigg, Ffion Coomber.

**Funding acquisition:** Alison Cooper, Natalie Joseph-Williams, Ruth Lewis, Adrian Edwards.

**Investigation:** Micaela Gal, Alison Cooper, Natalie Joseph-Williams, Elizabeth Doe, Ruth Lewis, Rebecca Jane Law, Sally Anstey, Nathan Davies, Amy Walters, Lisa Trigg, Steven Macey, Jane Greenwell, Ffion Coomber, Adrian Edwards.

**Methodology:** Micaela Gal, Alison Cooper, Natalie Joseph-Williams, Elizabeth Doe, Ruth Lewis, Rebecca Jane Law, Sally Anstey, Nathan Davies, Amy Walters, Robert Orford, Brendan Collins, Lisa Trigg, Chris Roberts, Steven Macey, Andrew Carson-Stevens, Jane Greenwell, Ffion Coomber, Adrian Edwards.

**Project administration:** Micaela Gal, Elizabeth Doe, Jane Greenwell, Ffion Coomber.

**Resources:** Micaela Gal, Alison Cooper, Natalie Joseph-Williams, Elizabeth Doe, Ruth Lewis, Rebecca Jane Law, Sally Anstey, Nathan Davies, Amy Walters, Jane Greenwell, Ffion Coomber, Adrian Edwards.

**Supervision:** Alison Cooper, Adrian Edwards.

**Validation:** Micaela Gal, Alison Cooper, Natalie Joseph-Williams, Elizabeth Doe, Ruth Lewis, Rebecca Jane Law, Sally Anstey, Nathan Davies, Amy Walters, Robert Orford, Brendan Collins, Lisa Trigg, Chris Roberts, Sarah Meredith, Steven Macey, Andrew Carson-Stevens, Jane Greenwell, Adrian Edwards.

**Visualization:** Micaela Gal, Alison Cooper, Natalie Joseph-Williams, Elizabeth Doe, Rebecca Jane Law, Sally Anstey, Nathan Davies, Amy Walters, Jane Greenwell, Ffion Coomber, Adrian Edwards.

**Writing – original draft:** Micaela Gal.

**Writing – review & editing:** Micaela Gal, Alison Cooper, Natalie Joseph-Williams, Elizabeth Doe, Ruth Lewis, Rebecca Jane Law, Sally Anstey, Nathan Davies, Amy Walters, Robert

Orford, Brendan Collins, Lisa Trigg, Chris Roberts, Sarah Meredith, Steven Macey, Andrew Carson-Stevens, Jane Greenwell, Ffion Coomber, Adrian Edwards.

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
