## [Decision Letter · Decision Letter 0]

6 May 2024

PONE-D-23-22096Knowledge mobilisation of rapid evidence reviews to inform health and social care policy and practice in a public health emergency: appraisal of the Wales COVID-19 Evidence Centre processes and impact, 2021-23PLOS ONE

Dear Dr. Cooper,  Thank you for submitting your manuscript to PLOS ONE. After careful consideration, we feel that it has merit but does not fully meet PLOS ONE’s publication criteria as it currently stands. Therefore, we invite you to submit a revised version of the manuscript that addresses the points raised during the review process. The insights shared in your manuscript are invaluable, and I appreciate the effort put into documenting this important work.

While the manuscript provides a comprehensive overview of the WCEC's activities, there are some areas where additional detail and clarification would enhance its readability and impact. I have summarized the key points from the reviewers' comments below:

Enhancement of Methods Section: Reviewers suggested providing more detail on stakeholder engagement, collaborative work processes, and data collection methods to offer readers a clearer understanding of the WCEC's approach.

Improvement in Presentation: Incorporating more visuals, simplifying language, and exploring alternative presentation methods such as videos or poems could make the information more accessible and engaging for a wider audience.

Streamlining of Content: Streamlining research questions and aligning rapid review processes with publication criteria were recommended to ensure clarity and manage expectations effectively.

Stakeholder Engagement: Continued close collaboration with stakeholders, including policy and practice decision-makers, was emphasized to enhance the utilization and impact of evidence reviews.

I believe that addressing these suggestions will significantly strengthen the manuscript and amplify its contribution to informing health and social care policy and practice during public health emergencies like the COVID-19 pandemic.

Additionally, I would like to propose the inclusion of an article featuring an interview with Prof. Marianna Arvanitakis, the winner of the European Award for the Best Medical Practice in the COVID-19 Pandemic. This article could highlight how recognition of exemplary practices can serve as a valuable resource in crisis situations and inform future health and social care policies. available here: https://globalbioethicsenquiry.com/wp-content/uploads/2023/07/VP2-JASNA.pdf

Thank you for considering these suggestions. I look forward to seeing the revised manuscript and the potential addition of the suggested article.

Best regards,

Reviewers' comments:

Reviewer's Responses to Questions

**Comments to the Author**

1. Is the manuscript technically sound, and do the data support the conclusions?

Reviewer #1: Partly

Reviewer #2: Partly

2. Has the statistical analysis been performed appropriately and rigorously? 

Reviewer #1: N/A

Reviewer #2: No

3. Have the authors made all data underlying the findings in their manuscript fully available?

Reviewer #1: No

Reviewer #2: No

4. Is the manuscript presented in an intelligible fashion and written in standard English?

Reviewer #1: Yes

Reviewer #2: Yes

5. Review Comments to the Author

Reviewer #1: Thank you for the opportunity to review this manuscript, and congratulations to the WCEC for the important work they completed during the COVID-19 pandemic. I appreciate how difficult these types of reports are to write, as they don't fit into the traditional IMRaD format of our typical scientific papers. Overall, I would like to have more detail within the methods section to better understand what steps the WCEC undertook as part of their work and KM. I have provided some specific examples of this below.

Background

- Page 4, Lines 100-102 - The authors make an important point that research evidence is not the only factor that decisions should be made on. I think it would be important to note that values/preferences (of individuals and communities) and also resources (financial, human, etc.) also play an important role in decision-making

- Page 4, Line 103 - this sentence is a bit awkward (acheiving impact is enabled...)

Figure 1: I like the use of the KTA framework, but it would be helpful to have more of the original steps maintained in the figure so the reader can see how the WCEC applied/adapted it for their KM purposes.

Methods

- It would be helpful for the authors to provide more detail on how the various stakeholder groups (Table 1) were engaged. How were questions identified and prioritized?

- I would like to read more about how the WCED and stakeholders worked collaboratively (page 7, Lines 155-162). Were three online meetings completed per review or over the course of the WCEC operations?

- What are some concrete examples of capacity building, sharing methodologies, and identifying gaps as part of the "associated activities" - maybe a table listing these would be helpful?

- Who were the stakeholders who were invited to take part in the survey? Were there any eligibility criteria set out? How were they invited to participate?

- What is meant by "accepted onto the work program" (page 13, line 282-4? Only these questions were taken on by the WCEC or only these questions were evaluated through the survey?

- More detail could be added as to how citations were identified, how email and meeting sessions were used to capture data when surveys were incomplete

Results

- What is meant by "good engagement" - how was this determined vs. poor engagement for example? (page 13, line 294).

- Did all stakeholders take part in all steps (lines 295-299)

- How many reviews were undertaken in total? How many were evaluated through the survey and how many through the other supplementary data collection methods?

- Can you provide demographic data for survey respondents?

- I believe the quotations come from the open ended questions in the survey. How was this qualitative data analysed? (it is not mentioned in the methods)

- SImilar comment on Page 14, Lines 320, what is meant by "worked well" - how was this determined? Two examples are given, but it may be helpful to have this information collated across all reviews and a summary presented

Table 2

- I now see that information on the 51 completed reviews is listed in Table 2 but I think that information should be a key feature in the results

- Timeline of completion - is it possible to give a mean or IQR rather than "usually 1 week to 3 months"

- The authors refer to different parts of the report, (topline summary, etc.). It would be helpful to have described earlier what a typical report included

- This table lists some examples of impact, but it would be more comprehensive if this was collated across reviews, rather than just choosing a few examples (which may be perceived as biasing the anticipated impact)

I'm not clear the distinction between survey results and the "feedback and collection of metrics" section 3.5.2 - how was this data collected if not through the survey?

Table 6 - how was this list compiled? Was a certain process used, or would this be better placed as part of the discussion? In fact, much of it is repeated in the discussion.

Minor suggestions for the authors' consideration

- There are a number of longer sentences, particularly in the background which could be broken up to enhance readability.

- Page 3, Line 88, spell out the acronym for BRIDGE criteria

Reviewer #2: This scoping review focuses on the role of care managers and the knowledge mobilization steps undertaken by the Welsh Centre for Evidence and Dissemination (WCEC) during the COVID-19 pandemic. The WCEC engaged closely with stakeholders, utilized the Knowledge to Action framework, and successfully disseminated evidence reviews to inform health and social care decision-making.

However, some significant concerns need to be addressed before publication.

1. Incorporate more infographics and visuals to make the information more accessible and engaging for a wider audience.

2. Explore different ways to present findings, such as creating poems or videos to cater to various learning preferences and increase engagement.

3. Enhance Language Accessibility: Simplify language in reports so that key points are clear to all stakeholders, including those with various levels of expertise.

4. Conduct evaluations on different dissemination methods to assess their impact and effectiveness in reaching and engaging stakeholders.

5. Streamline Questions: Formulate more specific and narrow research questions to ensure deliverability and manage expectations throughout the process.

6. Involve stakeholders in the knowledge mobilization plan from the beginning to gather valuable insights and ensure relevance.

7. Continue close collaboration with stakeholders, including public partnership members, to ensure that evidence outputs are targeted, timely, and effectively used.

8. Reflect on the impact requirements for academia and explore ways to align rapid reviews with the peer-reviewed journal publication criteria.

9. Develop more robust and long-term engagement strategies with policy and practice decision-makers to enhance the utilization and impact of evidence reviews.

10. Improve communication with stakeholders at all process stages to manage demand effectively and ensure expectation clarity.

6. PLOS authors have the option to publish the peer review history of their article (what does this mean?). If published, this will include your full peer review and any attached files.

Reviewer #1: No

Reviewer #2: No

---

## [Author Response · Author response to Decision Letter 0]

12 Jul 2024

Dear PLOS ONE Editor,

We wish to thank you for your time and effort in reviewing our manuscript so carefully and for providing the very helpful suggestions and comments .

We have done our best to address all the comments and suggestions of the editor and both reviewers, and believe that this has significantly strengthened the manuscript, and amplified its contribution to informing health and social care policy and practice during public health emergencies like the COVID-19 pandemic. 

The explanations on how we have addressed the editor suggestions and comments are provided below.

Thank you for your time and consideration of the revised manuscript.

Yours sincerely, Micaela Gal

(Note: the references to pages and line numbers given below, refer to the ‘marked up’ version of the manuscript (named: Revised article with changes highlighted)).

1. Editor suggestion: Enhancement of Methods Section: Reviewers suggested providing more detail on stakeholder engagement, collaborative work processes, and data collection methods to offer readers a clearer understanding of the WCEC's approach.

Author response:

We agree that more detail is needed in the methods section related to stakeholder engagement, collaborative work processes and data collection. 

We have included more information on each of the suggestions for improving the methods, and provide evidence of this in our responses to Reviewer 1.

We have also added 4 additional references (References 18-21) from the Evidence Centre, which cover in-depth the overall approach used by the Centre for stakeholder identification and engagement, identification and prioritisation of the research questions, and how the stakeholders and evidence centre team worked together collaboratively during the evidence review process. 

2. Improvement in Presentation: Incorporating more visuals, simplifying language, and exploring alternative presentation methods such as videos or poems could make the information more accessible and engaging for a wider audience.

Author response:

Incorporating more visuals:

We have included 2 additional tables and 2 figures to enhance and explain the text. These are in line with the comments from the reviewers:

Table 2. Information contained within the front pages of the Centre reports (page 11)

Table 5. Examples of associated activities conducted by the WCEC (page 22)

Fig 2. Questions received, prioritised, and accepted onto the Centre work programme (page 9)

Fig 4. Evidence Centre rapid review (evidence synthesis) outputs (page 20)

Simplifying language:

We have reviewed the manuscript and simplified the language used, to make it more accessible. 

Alternative presentation methods:

The suggestions to use alternative methods of presentation such as videos or poems relate to Feedback provided by Reviewer 2. This was helpful and insightful. Indeed, one of our public group members had suggested poetry for one of our Evidence Centre reports, which focused on children and education during the pandemic. We have now included this in the discussion (page 31 lines 590 -595):

A suggestion for wider dissemination of findings in a different format was made by a public member (e.g., review findings relevant to children and young people could be made into a poem). We acknowledge that wider dissemination methods such as creating stories, poems or videos would likely be valuable to make findings more accessible and engaging to wider groups. However, this was not possible at the time due to resource constraints (available staff time and additional funding). 

We have also included an additional reference (*40), which refers readers to a public facing legacy report for the Wales COVID-19 Evidence Centre (*Wales COVID‐19 Evidence Centre Legacy Report. Wales COVID-19 Evidence Centre. June 2023. https://researchwalesevidencecentre.co.uk/sites/default/files/2023-11/Wales_COVID-19_Evidence_Centre_Legacy_Report_June23_ENGLISH.pdf)

We have included the following in our Discussion section (page 28, lines 567 to 571):

To make the work of the WCEC more widely accessible, the Centre also produced a publicly available legacy report.(40) This report includes information about the Centre structure, methods, reviews, knowledge mobilisation and impact. The lay summaries accompanying each review were written by members of the Centre public partnership group, and together with the infographics, increase accessibility of the review findings.

3. Editor suggestion: Streamlining of Content: (i) Streamlining research questions and (ii) aligning rapid review processes with publication criteria were recommended to ensure clarity and manage expectations effectively.

Author response:

i. Streamlining research questions:

The research questions, which the Centre received from the stakeholders were typically very broad. The Centre had to work with the stakeholders to manage their expectations, and streamline the questions to ensure that the reviews could deliver useful information and be completed within the stakeholder’s timeframe. The streamlining/focusing of the questions was part of the rapid evidence review process, and not part of the knowledge mobilisation processes of the Centre, which were the focus of this manuscript (and so was not described). 

However, we agree that this point would be of interest to readers of this manuscript. Therefore, we have made reference to this in the Discussion section (Page 28, Lines 523 -525), and included a reference to another Centre manuscript, where the streamlining of research questions for the Evidence reviews is described in detail (below):

The responsiveness and involvement of stakeholders and members of the public were key to focusing the research questions, which were usually too broad to be answered in the available amount of time.(23)

A bespoke rapid evidence review process engaging stakeholders for supporting evolving and time-sensitive policy and clinical decision-making: reflection and lessons learned from the Wales Covid-19 Evidence Centre 2021-23. Lewis R et al. https://assets-eu.researchsquare.com/files/rs-3878814/v1/59d98322-ac9e-492b-aead-afbd02db0b37.pdf?c=1706010939

ii. Aligning rapid review processes with publication criteria:

We acknowledge that this is an interesting point and important point. 

Our rapid review processes including the methodology and report content are aligned to approved methods for conducting rapid reviews. The findings from a number of the rapid reviews have now been published in peer reviewed journals and met the publication criteria needed for these. 

We have now made reference to this in the discussion (Page 35, Table 8, last box):

When producing rapid reviews, the requirements and criteria of peer reviewed journals and the research and impact requirements of academia should be considered. Our review methodology is aligned to recognised rapid review methodology and a number of the reviews have now been published in peer reviewed journals.(23)

The method for our rapid review process is also described in a referenced manuscript, which has now been added to our manuscript:

A bespoke rapid evidence review process engaging stakeholders for supporting evolving and time-sensitive policy and clinical decision-making: reflection and lessons learned from the Wales Covid-19 Evidence Centre 2021-23. Lewis R et al. https://assets-eu.researchsquare.com/files/rs-3878814/v1/59d98322-ac9e-492b-aead-afbd02db0b37.pdf?c=1706010939

4. Editor suggestion: Stakeholder Engagement: Continued close collaboration with stakeholders, including policy and practice decision-makers, was emphasized to enhance the utilization and impact of evidence reviews.

Author response: 

We had already included some information regarding this in the Discussion (page 28, lines 529-536). We have included an additional paragraph this in the manuscript Discussion (page 29 line 537-541).

Following the end of funding of the Wales COVID-19 Evidence Centre, we received 5 further years of funding, and are now funded as the ‘Health and Care Research Wales Evidence Centre (2023-2028). This enables us to continue close collaborations with the stakeholders we have worked with in the COVID-19 Evidence Centre to enhance use of our report findings, and collect and evidence longer term evidence of impact

5. Editor suggestion: Additionally, I would like to propose the inclusion of an article featuring an interview with Prof. Marianna Arvanitakis, the winner of the European Award for the Best Medical Practice in the COVID-19 Pandemic. This article could highlight how recognition of exemplary practices can serve as a valuable resource in crisis situations and inform future health and social care policies. available here: https://globalbioethicsenquiry.com/wp-content/uploads/2023/07/VP2-JASNA.pdf

Author response:

We agree that highlighting how examples of exemplary practice during the pandemic should serve as valuable resources and points of learning for future crisis situations, as well as informing health and social care policies going forward. We have made reference to this and added the citation (Page 3, Lines 78-81)

There are some valuable examples of exemplary practices during the pandemic. These should serve as a valuable resource and lessons for managing future crisis situations, and to inform future health and social care policies.(3)

5. Editor suggestion: Please ensure that your manuscript meets PLOS ONE's style requirements, including those for file naming.

Author response:

We have ensured that the revised manuscript meets PLOS ONE style requirements, including for the naming of the files. 

We have also used the Preflight Analysis and Conversion Engine (PACE) digital diagnostic tool, to ensure that figures meet PLOS ONE style requirements.

6. Editor suggestion: Data sharing for the study. 2. We note that you have indicated that there are restrictions to data sharing for this study. For studies involving human research participant data or other sensitive data, we encourage authors to share de-identified or anonymized data. However, when data cannot be publicly shared for ethical reasons, we allow authors to make their data sets available upon request. Before we proceed with your manuscript, please address…

Author response:

We agree that research data should be made available wherever possible. 

The anonymised survey data (questions and responses (answer selection and free text) from the Stakeholder survey has now been made avaialble on the repository Figshare (https://figshare.com). The data file can be downloaded here: 

https://figshare.com/articles/dataset/Wales_COVID-19_Evidence_Centre_2021-2023_Stakeholder_survey/25912459

Some data and identifiers were removed before adding the data to Figshare (i.e. names. job title, place of work, comments that may identify participants, feedback where participants did not give permission to quote this). This was done to ensure the data was anonymised, as respondents had not consented to have this data shared without anonymisation. 

We have included this information in the Results section – 3.5.1 Stakeholder survey (Page 24, Line 473-474): 

The data from the Stakeholder survey are accessible on the data repository Figshare (Wales COVID-19 Evidence Centre 2021-2023 Stakeholder survey)

The Data Availability statement in the submission form has been updated accordingly.

We have shown how we have addressed Reviewer 1 and Reviewer 2 suggestions and comments detail in our 'Response to reviewers' document, which has been uploaded.

---

## [Decision Letter · Decision Letter 1]

10 Sep 2024

PONE-D-23-22096R1Knowledge mobilisation of rapid evidence reviews to inform health and social care policy and practice in a public health emergency: appraisal of the Wales COVID-19 Evidence Centre processes and impact, 2021-23PLOS ONE

Dear Dr. Gal,

Thank you for submitting your manuscript to PLOS ONE. After careful consideration, we feel that it has merit but does not fully meet PLOS ONE’s publication criteria as it currently stands. Therefore, we invite you to submit a revised version of the manuscript that addresses the points raised during the review process.

Further to your previous revisions, Reviewer 1 has requested/re-requested some changes to improve the clarity and reporting of your study in this manuscript. Please revise carefully and ensure each comment is addressed in the response to reviewers document submitted with your revised manuscript. 

We look forward to receiving your revised manuscript.

Kind regards,

Jennifer Tucker, PhD

Staff Editor

PLOS ONE

Reviewers' comments:

Reviewer's Responses to Questions

**Comments to the Author**

1. If the authors have adequately addressed your comments raised in a previous round of review and you feel that this manuscript is now acceptable for publication, you may indicate that here to bypass the “Comments to the Author” section, enter your conflict of interest statement in the “Confidential to Editor” section, and submit your "Accept" recommendation.

Reviewer #1: (No Response)

2. Is the manuscript technically sound, and do the data support the conclusions?

Reviewer #1: Partly

3. Has the statistical analysis been performed appropriately and rigorously? 

Reviewer #1: N/A

4. Have the authors made all data underlying the findings in their manuscript fully available?

Reviewer #1: No

5. Is the manuscript presented in an intelligible fashion and written in standard English?

Reviewer #1: Yes

6. Review Comments to the Author

Reviewer #1: Thank you to the authors for their revisisions, the manuscript is much improved.

I have a few additional/outstanding comments.

1. Was ethical approval sought or received for this survey? It is not mentioned anywhere

2. There are some inconsistencies in the use of acronyms throughout (e.g., TAC on line 160 and spelled out on line 165).

3. Line 172 - were questions received and prioritized at one time or on a rolling basis? I am guessing the later but it is not clear.

4. Line 182 - what is the collaborating partner review team?

5. Methods - I'm not sure if collaborating and co-production are the appropriate terms based on what is described (which is understandable given the very tight timelines the team was working on). For example, using the IAP2 framework for engagement, what the authors describe sounds more like "consult" or maybe "involve". Perhaps a definition of co-production and collaboration with citation would be helpful.

6. Section 2.4 - the authors mention gaps were "collated" - how was this done, and what was done with this information? are there any examples you can provide?

7. Line 348 seems to be missing something ("agreeing relevant outcomes")

8. The authors define good vs. poor engagement, but this probably fits better in the methods (if it was prespecified) vs. the results

9. Line 359 - n = 21 responded to the survey, how many were invited?

10. The authors have provided more information as to how the qualitative data were collected, but there is no description of data analysis approaches for the qualitative (or quantitative data for that matter)

11. Line 439 - how many of the 21 surveys were incomplete?

12. Table 7 could be integrated with Table 4, perhaps as a row of "policy impacts"; there is already some duplication in the table (e.g., infographics for midwives).

13. With respect to facilitators to the impact or how to better mobilize next time, where did the resources come from to develop this centre?

14. Table 8 - i'm unclear if these are part of the "results" (under the objective to provide recommendations for best practice" or discussion. Perhaps if the authors were more clear on whether these are personal reflections or more informed by data.

15. The strengths/limitations section should focus more on the strengths/limitations of this paper and its evaluation rather than the centre as a whole.

16. Another pass for readability is advised... for example "it is worth noting" is on line 544 and line 551.

7. PLOS authors have the option to publish the peer review history of their article (what does this mean?). If published, this will include your full peer review and any attached files.

Reviewer #1: No

---

## [Author Response · Author response to Decision Letter 1]

23 Oct 2024

We have uploaded a document 'Response to Editor and Reviewer comments_ October 2024', where we indicate how we have addressed all editor and reviewer comments in detail. This also documents where resulting changes have been made in the manuscript.

---

## [Decision Letter · Decision Letter 2]

12 Nov 2024

Knowledge mobilisation of rapid evidence reviews to inform health and social care policy and practice in a public health emergency: appraisal of the Wales COVID-19 Evidence Centre processes and impact, 2021-23

PONE-D-23-22096R2

Dear Dr. Gal,

We’re pleased to inform you that your manuscript has been judged scientifically suitable for publication and will be formally accepted for publication once it meets all outstanding technical requirements.

Kind regards,

Sreeram V. Ramagopalan

Academic Editor

PLOS ONE

Additional Editor Comments (optional):

Reviewers' comments:

Reviewer's Responses to Questions

**Comments to the Author**

1. If the authors have adequately addressed your comments raised in a previous round of review and you feel that this manuscript is now acceptable for publication, you may indicate that here to bypass the “Comments to the Author” section, enter your conflict of interest statement in the “Confidential to Editor” section, and submit your "Accept" recommendation.

Reviewer #1: All comments have been addressed

2. Is the manuscript technically sound, and do the data support the conclusions?

Reviewer #1: Yes

3. Has the statistical analysis been performed appropriately and rigorously? 

Reviewer #1: N/A

4. Have the authors made all data underlying the findings in their manuscript fully available?

Reviewer #1: Yes

5. Is the manuscript presented in an intelligible fashion and written in standard English?

Reviewer #1: Yes

6. Review Comments to the Author

Reviewer #1: The authors have adequately responded to all of the comments provided. I look forward to seeing this paper in print.

7. PLOS authors have the option to publish the peer review history of their article (what does this mean?). If published, this will include your full peer review and any attached files.

Reviewer #1: No

---

## [Editor Report · Acceptance letter]

15 Nov 2024

PONE-D-23-22096R2 

PLOS ONE

Dear Dr. Gal, 

I'm pleased to inform you that your manuscript has been deemed suitable for publication in PLOS ONE. Congratulations! Your manuscript is now being handed over to our production team.

Kind regards, 

on behalf of

Dr. Sreeram V. Ramagopalan 

Academic Editor

PLOS ONE